


**A machine learning approach targeting parameter estimation for plant**
**functional type coexistence modeling using ELM-FATES (v2.0)**
Lingcheng Li[1], Yilin Fang[2], Zhonghua Zheng[3], Mingjie Shi[1], Marcos Longo[4], Charles D. Koven[4],
Jennifer A. Holm[4], Rosie A. Fisher[5], Nate G. McDowell[1,6], Jeffrey Chambers[4], L. Ruby Leung[1]
1.  Atmospheric Sciences and Global Change Division, Pacific Northwest National Laboratory,
Richland, WA, USA
2.   Earth System and Science Division, Pacific Northwest National Laboratory, Richland, WA,
USA
3.  Department of Earth and Environmental Sciences, The University of Manchester, Manchester,
UK
4.  Climate and Ecosystem Sciences Division, Lawrence Berkeley National Laboratory, Berkeley,
CA, USA
5.  CICERO Center for International Climate and Environmental Research, Oslo, Norway
6.  School of Biological Sciences, Washington State University, PO Box 644236, Pullman, WA,
USA
Correspondence to: Lingcheng Li (lingcheng.li@pnnl.gov)



**Highlight**
• Machine learning based surrogate models were developed and used to optimize the
selection of the trait parameters in ELM-FATES demographic vegetation model
• Trait parameters selected by the surrogate models significantly improve the modeling of
plant functional type coexistence and reduce model errors.
• This approach represents a repeatable method for identifying parameter values that
satisfy fidelity against observations and coexistence between functional types in
vegetation demography models.





**Abstract**
Tropical forest dynamics play crucial roles in the global carbon, water, and energy cycles.
Dynamic global vegetation models are the primary tools to simulate terrestrial ecosystem
dynamics and their response to climate change. However, realistically simulating the dynamics of
competition and coexistence of differing plant functional traits within tropical forests remains a
significant challenge. This study aims to improve the modeling of plant functional type (PFT)
coexistence in the Functionally Assembled Terrestrial Ecosystem Simulator (FATES), a
vegetation demography model implemented in the Energy Exascale Earth System Model (E3SM)
land model (ELM), ELM-FATES. Specifically, we explore: (1) whether plant trait relationships
established from field measurements can constrain ELM-FATES simulations; and (2) whether
machine learning based surrogate models can emulate the complex ELM-FATES model and
optimize parameter selections to improve PFT coexistence modeling. We conducted ELM-FATES
experiments for a tropical forest site near Manaus, Brazil. We first conducted two ensembles of
ELM-FATES experiments, without (Exp-1) and with (Exp-2) consideration of observed trait
relationships, respectively. Considering the observed trait relationships (Exp-2) slightly improves
ELM-FATES simulations of water, energy, and carbon fluxes, but degrades the simulation of PFT
coexistence. Using eXtreme Gradient Boosting (XGBoost) based surrogate models trained on Exp-
1, we optimize the trait-related parameters in ELM-FATES to enable PFT coexistence and reduce
model errors relative to the field observations. We used parameters selected by the surrogate model
to conduct another ensemble of ELM-FATES experiments (Exp-3). The probability of
experiments yielding PFT coexistence greatly increases from 21% in Exp-1 to 73% in Exp-3.
Further filtering those experiments that allow for PFT coexistence to agree within 15% of the
observations, Exp-3 still has 33% of experiments left, much higher than the 1.4% in Exp-1. Exp-
3 also better reproduces the annual means and seasonal variations of water, energy and carbon
fluxes, and the field inventory of above ground biomass. Our study demonstrates the benefits of
using machine learning models to improve PFT coexistence modeling in ELM-FATES, with
important implications for modeling the response and feedback of ecosystem dynamics to climate
change. Our results also suggest that new mechanisms are required for robust simulation of
coexisting plants in FATES.



**Plain Language Summary**
Modeling tropical forest dynamics is crucial for understanding global carbon, water, and energy
cycles under climate change. Dynamic global vegetation models, the primary tools to simulate
terrestrial ecosystem dynamics, face the challenge of realistically modeling the competition and
coexistence of different plant functional types (PFT). Our study explores whether (1) using plant
trait measurements and (2) developing machine learning based surrogate models to optimize
parameter selections can improve plant coexistence modeling. Using ELM-FATES as a testbed,
multiple ensembles of numerical experiments are conducted for a tropical forest site. We found
there is limited guidance of observed trait relationships for PFT coexistence modeling in ELM-
FATES. Trait parameters selected by the surrogate models significantly improve the modeling of
PFT coexistence and reduce model errors. We demonstrate the benefits of developing machine
learning based surrogate models to improve PFT coexistence modeling in ELM-FATES, with
important implications for modeling the response and feedback of ecosystem dynamics to climate
change. Our results also suggest that new mechanisms are required for robust simulation of
coexisting plants in ELM-FATES.



## 1. Introduction

Tropical ecosystems feature the highest biodiversity on Earth, maintaining more than 75% of all known species (Mora et al., 2011; Mitchard, 2018). The dynamics of tropical forests are closely related to the regional and global carbon, energy and water cycles (Bonan, 2008; Piao et al., 2020). Vegetation is expected to face more water stress from vapor pressure deficit increase and soil moisture reduction with global warming (McDowell et al., 2020). Forest dynamics of tree mortality are accelerating in some tropical regions due to the rising atmospheric water stress (Bauman et al., 2022; Hubau et al., 2020; Zuleta et al., 2017). Tropical forests currently make an approximately neutral contribution to the global carbon cycle as a result of a large land-use source balanced by sinks in recovering and undisturbed forests, but they may become a carbon source in the future under the threat of climate change and human-induced disturbance (Mitchard, 2018; Gatti et al., 2021). Therefore, understanding and modeling tropical forest dynamics and related feedbacks have crucial implications for projecting future changes in the global climate system.

Dynamic global vegetation models (DGVMs) are the primary tools to simulate terrestrial ecosystem dynamics of plant functional type distribution, ecosystem composition and functioning, and ecosystem response to and recovery from disturbance (e.g., fire and wind damage) (Longo et al., 2019; Fisher et al., 2018; Foley et al., 1996; Sitch et al., 2003; Cao and Woodward, 1998; Berzaghi et al., 2019; McMahon et al., 2011). Conventional DGVMs represent plant communities using an area-averaged representation of plant functional types (PFTs) in each grid cell. Their relatively simple structures have the advantage of high computational efficiency for use in Earth system models (Fisher et al., 2018; Snell et al., 2014). However, these models do not capture many demographic processes. For example, plants of each represented PFT typically have identical



properties (e.g., tree size), which limits the capability of modeling ecosystem dynamics and
functioning of canopy gap formation, PFT competition, and disturbance reactions (Feeley et al.,
2007; Stark et al., 2012; Hurtt et al., 1998; Moorcroft, 2003; Brister et al., 2020). To overcome
these limitations, individual-based models, also known as forest gap models, explicitly represent
vegetation as individual plants and simulate their birth, growth, and death (Fyllas et al., 2014;
Christoffersen et al., 2016; Sato et al., 2007; Jonard et al., 2020; Maréchaux and Chave, 2017).
These models incorporate the stochasticity and heterogeneity of the plant light environment
mechanistically and thereby can typically represent PFT competitive exclusion, succession, and
coexistence. However, explicit simulations of individual plants with stochastic processes suffer a
substantial computational penalty and limit applicability over large or global scales (Fisher et al.,
2018). To capture sufficient ecosystem dynamics and maintain relatively high computational
efficiency, "cohort-based" models have been proposed (Haverd et al., 2013; Medvigy et al., 2009;
Ma et al., 2021; Moorcroft et al., 2001; Longo et al., 2019). In a cohort-based approach, individual
plants are grouped together as "cohorts" based on their similar properties, including size, age, and
PFT (Fisher et al., 2018). Many cohort-based models have been developed and widely used across
regional to global scales. Examples of cohort-based models include the Ecosystem Demography
model (ED) (Moorcroft et al., 2001), the Functionally Assembled Terrestrial Ecosystem Simulator
(FATES) (Fisher et al., 2018, 2015), and the Geophysical Fluid Dynamics Laboratory (GFDL)
Land Model 3 with the Perfect Plasticity Approximation (LM3-PPA) (Weng et al., 2015). Among
these models, FATES has been widely used in modeling ecosystem dynamics for multiple
ecosystems, e.g., tropical (Holm et al., 2020; Koven et al., 2020; Cheng et al., 2021) and mixed-
conifer forests (Buotte et al., 2021), and forest disturbance (Huang et al., 2020).



Despite ongoing applications, robust simulations of competition and coexistence in cohort-based
DGVMs remain a major challenge. In niche-based coexistence theory, coexisting species require
both convergence in strategy to adapt to the surrounding environment ("environmental filtering")
and divergence in strategy to ensure differentiation in resource requirements ("niche partitioning")
(Kraft et al., 2008; Adler et al., 2013). These same constraints apply to coexisting PFTs as modeled
by DGVMs. Thus, on the one hand, DGVMs need to include mechanisms involving critical niche
dimensions (e.g., light, water, and nutrients). For example, the multi-layer canopy structure in
FATES provides vertical light resource differentiation. Another essential aspect is to assign
reasonable plant functional traits (i.e., the parameters that define a given plant functional type) to
satisfy environmental filtering, ensure niche partitioning, and consequently preserve PFT
coexistence. Considering the relatively high computational cost of DGVMs and the host land
surface models, it is not feasible to directly apply global optimization methods such as Shuffled
Complex Evolution (Duan et al., 1992) to calibrate trait-related parameters, because this could be
time-consuming and computationally intensive (Rouholahnejad et al., 2012). Therefore, most
previous studies use the filtered ensemble approach to select trait-related parameters involving
several steps: 1) generate a parameter ensemble based on reference trait ranges or correlations, 2)
conduct ensemble model simulations, and 3) filter the parameter ensemble by coexistence and
other criteria (e.g., observation constraints). For example, Huang et al. (2020) applied FATES
implemented in the Community Land Model (CLM; herein CLM-FATES) with two tropical PFTs
to study forest dynamics at tropical sites. They performed 70 one-at-a-time experiments before
obtaining one reasonable parameter set. Buottte et al. (2021) used CLM-FATES to simulate forest
dynamics of pine and incense cedar over the Sierra Nevada of California, and their two stages of
experiments (360 plus 72 runs) only yielded four sets of parameters that met the given criteria. The



filtered ensemble approach has low efficiency, which hinders DGVMs' application to modeling
ecosystem dynamics under the changing climate. In addition, trait relationships derived from field
measurements are often used to infer parameter selections when simulating coexistence. For
example, Longo et al. (2020) used multiple trait relationships derived from various datasets to
guide parameter selection for different PFTs in the ED-2.2 model simulations. However, whether
the observed trait relationships can efficiently improve PFT coexistence simulation in current
DGVMs is still unclear.

Machine learning (ML) has facilitated Earth science studies (Shen, 2018; Nearing et al., 2021; Zhu
et al., 2022; Pal et al., 2019; Jung et al., 2019), possibly providing a promising approach to improve
PFT coexistence modeling in DGVMs. ML algorithms have been broadly and successfully
employed in recent decades. They can be used as standalone models to predict variables of interest
or integrated with process-based models to improve simulations from the latter (Xu and Liang,
2021; He et al., 2022). Among these applications, ML has shown advantages as a surrogate model
for parameter optimization and sensitivity quantification, including its effectiveness and easy
application, its ability to implicitly deal with complex nonlinear correlations and high dimensional
data, and handle interactions between variables (Sit et al., 2020; Antoniadis et al., 2020; Tsai et al.,
2021). One promising approach is to construct ML-based surrogate models using data from initial
model simulations to emulate the relationship between inputs (i.e., model parameters) and model
outputs (Wang et al., 2014). Then the computationally inexpensive surrogate model can be
efficiently used for parameter optimization and sensitivity analysis. For example, Dagon et al.
(2020) implemented artificial neural networks to emulate the satellite leaf area constrained version
of CLM5 (Lawrence et al., 2019) and estimated optimal parameters to improve the global



simulation of gross primary production and latent heat flux. Sawada (2020) developed an ML
surrogate model to optimize the land surface model parameters and improve soil moisture and
vegetation dynamics simulations. Watson-Parris et al. (2021) built a general tool to efficiently
emulate Earth system models for uncertainty quantification and model calibration. Although
employing ML based surrogate models to optimize the trait parameters and hence improve the
vegetation dynamics modeling in DGVMs is promising, this area of research is still under-explored.

This study aims to improve PFT coexistence modeling in DGVMs. The cohort-based FATES
implemented in the Energy Exascale Earth System Model (E3SM) land model (ELM; Golaz et al.,
2019), i.e., ELM-FATES, is taken as our testbed. The ELM land model simulates surface energy
fluxes, soil and canopy biophysics, hydrology, and soil biogeochemistry, whereas FATES
simulates live vegetation processes, litter dynamics, and fire. We first examine whether trait
relationships constructed from field measurements can help improve ELM-FATES simulations.
Second, we explore whether ML based surrogate models can help optimize key trait parameters in
ELM-FATES to improve the simulation of PFTs coexistence. Our model experiments are
conducted for a tropical rainforest site located in Manaus, Brazil. This paper is organized as
follows. Section 2 describes ELM-FATES, summarizes the key functional trait-related parameters,
introduces the machine learning algorithms, and explains the overall experimental design. Results
are presented in Section 3, followed by Discussions and Conclusions in Section 4 and Section 5,
respectively.



**185**    **2. Methodology**

**186**    **2.1 Study site and data**

**187**    Our study site is located at kilometer 34 (K34) of the ZF2 road, Manaus, Brazil (latitude: -2.6091

**188**    S; longitude: -60.2093 W). The K34 site is an old-growth primary forest with minimal human

**189**    disturbances (Holm et al., 2020). The annual precipitation is about 2252 mm, and the mean

**190**    temperature is about 26.68 °C (https://ameriflux.lbl.gov/sites/siteinfo/BR-Ma2). The wet season is

**191**    from November to May, and the dry season is from June to October (Fang et al., 2017). Hourly

**192**    meteorological forcing (i.e., precipitation, air temperature, relative humidity, wind speed, surface

**193**    pressure) at the K34 eddy covariance flux tower from 2002–2005 was obtained from the LBA-

**194**    ECO CD-32 Flux Tower Network Data Compilation (Restrepo-Coupe et al., 2021). Observational

**195**    reference datasets obtained from Holm et al. (2020) include gross primary production (GPP),

**196**    evapotranspiration (ET), sensible heat flux (SH), Bowen ratio (BW, the ratio between sensible heat

**197**    and latent heat), and inventory data-based aboveground biomass (AGB). The GPP, ET, SH, and

**198**    BW observations are monthly climatological averages from 2000 to 2008 (Table S1). The AGB at

**199**    this site is about $303 \pm 2.3$ Mg/ha. These observational data were used to evaluate the ELM-

**200**    FATES simulations and constrain the ML surrogate models.

**201**

**202**    **2.2 ELM-FATES and parameters**

**203**    ELM-FATES is used as the testbed. ELM is the land model of E3SM, which is the host land model

**204**    of FATES (Golaz et al., 2019; Leung et al., 2020; Holm et al., 2020). FATES is a size- and age-

**205**    structured vegetation model developed from the Community Land Model with ecosystem

**206**    demography (CLM-ED) (Fisher et al., 2015; Koven et al., 2020). FATES includes two key

**207**    structural components: ecosystem demography (ED; Moorcroft et al., 2001) and a modified





version of perfect plasticity approximation (PPA, Purves et al., 2008). FATES discretizes the
simulated landscape into spatially implicit "*patches*" representing different disturbance histories
of the ecosystem since the last disturbance. Within each patch, the hypothetical population of
plants is grouped into "*cohorts*": a cohort consists of a population density of trees with similar size
and the same plant functional type. Cohorts are organized, via the PPA concept, into canopy layers,
and compete for light based on their canopy vertical positions (e.g., canopy layer vs. understory
layer). The understory layer is formed when the canopy area becomes greater than the total ground
area, and some fraction of each cohort is 'demoted' to the understory as a function of its height.
The number of patches and cohorts varies depending on processes, including recruitment, growth,
mortality, competition, and disturbance. The modified PPA probabilistically splits cohorts into
discrete canopy and understory layers based on a function of their height (Strigul et al., 2008;
Fisher et al., 2010). A detailed description of the FATES model can be found in its technical note
(Zenodo, https://doi.org/10.5281/zenodo.3517272).

In this study, we configured two PFTs in ELM-FATES, i.e., early successional and late
successional broadleaf evergreen tropical trees, which can represent a primary axis of variability
in tropical forests (Huang et al., 2020; Reich, 2014; Díaz et al., 2016). There are tradeoffs between
the plant traits of these two PFTs. Compared with the late successional PFT, the early successional
PFT is more light-demanding and fast-growing, but with lower woody density, shorter leaf and
root lifespans, and higher background mortality. To represent the drought impacts on forest
dynamics, the early successional PFT is further assumed to be less drought resistant with shallower
rooting depth and hence more easily affected by drought conditions (Oliveira et al., 2021). The



corresponding tradeoffs and parameters between these two PFTs are shown in Figure 1 and Table

231    1.

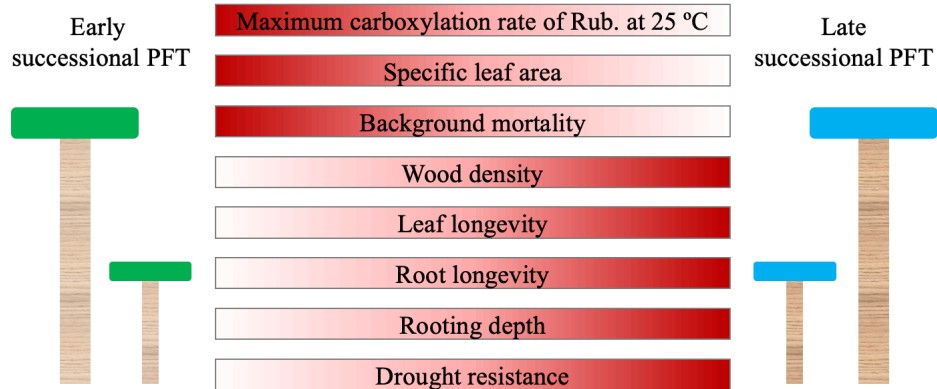


Figure 1. Schematic representation of tradeoffs between early and late successional PFTs. Dark
red denotes a higher parameter value. The tradeoffs of the top five traits are used to constrain the
parameter sampling.

Observational datasets have shown some correlations between plant traits. Therefore, we derived
three trait relationships based on the tropical studies of Koven et al. (2020) and Longo et al. (2020).
Using the digitized data from Figure 3 in Koven et al. (2020), background mortality $M_{bk}$ (see table
1 for parameter definitions) can be empirically computed from $V_{cmax}$,
$$M_{bk} = 0.0082 \times e^{(0.0153 \times V_{cmax})} \tag{1}$$
Based on the equations in Figure S18 of Longo et al. (2020), $L_{leaf}$ and $WD$ can be calculated via
$SLA$,
$$L_{leaf} = 0.0001 \times SLA^{(-2.32)} \tag{2}$$
$$WD = -0.583 \times \ln(SLA) - 1.6754 \tag{3}$$



These trait relationships are used in parameter generation to test whether considering trait
relationships can help ELM-FATES to model PFT coexistence.

249                     Table 1 Summary of ELM-FATES parameters for two PFTs

| Parameter type | Parameter name | Symbol | Unit | Early PFT | Late PFT | Range |
|---|---|---|---|---|---|---|
| Optimized parameter | Maximum carboxylation rate of Rub. at 25 ºC, canopy top | $V_{cmax}$ | µmol $CO_2$/m²/s | $V_{cmax,early} > V_{cmax,late}$ | | 40–105 |
| | Specific leaf area, canopy top | $SLA$ | m²/gC | $SLA_{early} > SLA_{late}$ | | 0.005–0.04 |
| | Background mortality rate | $M_{bk}$ | 1/yr | $M_{bk,early} > M_{bk,late}$ | | 0.005–0.05 |
| | Wood density | $WD$ | g/cm³ | $WD_{early} < WD_{late}$ | | 0.2–1.0 |
| | Leaf longevity | $L_{leaf}$ | year | $L_{leaf,early} < L_{leaf,late}$ | | 0.2–3.0 |
| | Maximum size of storage C pool, relative to the maximum size of leaf C pool | $CR_{s2l}$ | — | same | | 0.8–1.5 |
| Fixed parameter | Root longevity | $L_{root}$ | year | 0.9 | 2.6 | — |
| | Fine rooting distribution profile parameter a | $R_a$ | — | 7 | 7 | — |
| | Fine rooting distribution profile parameter b | $R_b$ | — | 2 | 0.4 | — |
| | BTRAN threshold below which drought mortality begins. | $M_{btran}$ | — | 0.4 | 1.0E-06 | — |
| | Soil water potential at full stomatal closure | $\psi_{closure}$ | mm | –113000 | –242000 | — |

*Parameter references (Huang et al., 2020; Koven et al., 2020; Longo et al., 2020; Holm et al., 2020; Cheng et al., 2021; Domingues et al.,
2005; Chitra-Tarak et al., 2021; Buotte et al., 2021)
*$R_a$ and $R_b$ are parameters that determine the rooting depth and vertical distribution of fine roots.
*BTRAN is the plant water stress factor. BTRAN ∈ [0,1], 0 representing full water stress, 1 representing no water stress.

**2.3 XGboost and SHAP**
In this study, we built ML-based surrogate models to emulate ELM-FATES simulations. To
represent the nonlinear relationship between ELM-FATES parameters and the model outputs (e.g.,
ET), we used eXtreme Gradient Boosting (XGBoost; Chen and Guestrin, 2016), a decision-tree-
based ensemble machine learning algorithm. The boosting algorithm sequentially trains a set of
weak learners (e.g., decision trees) to the ensemble, with each successive learner correcting the
biases/mistakes of its predecessors. XGBoost is a highly efficient and scalable algorithm built on
the Gradient Boosting framework (Friedman, 2001). For instance, it not only handles complex



nonlinear interactions and collinearity between different features (due to the decision tree's nature),
but also provides a parallel implementation that effectively solves a range of data science problems.
XGBoost has been successfully applied in a variety of fields within Earth and Environmental
Sciences, such as urban temperature emulation (Zheng et al., 2021b), wildfire burned area (Wang
et al., 2021), and emissions prediction (Wang et al., 2022), flash flood risk assessment (Ma et al.,
2021), and aerosol property estimation (Zheng et al., 2021a, c).

We performed parameter sensitivity analysis to understand which trait parameters are essential for
ELM-FATES simulations. A game theoretic approach called SHapley Additive exPlanations
(SHAP; Lundberg and Lee, 2017; Lundberg et al., 2018, 2020) was used to interpret the trained
XGBoost models and identify the relative importance of features. This approach assumes that
features (predictive variables) interact to participate in a game of prediction. The features receive
a payout for their contributions as a result of this collaboration. Compared to the intrinsic feature
importance methods (for example, feature importance in XGBoost), SHAP uses a unified measure
of feature importance to explain both individual samples and the entire dataset (Lundberg and Lee,
2017). This novel approach has been used to interpret a digital soil mapping model (Padarian et
al., 2020) and identify the critical drivers of wildfires (Wang et al., 2021). Specifically, we
performed SHAP analysis for each XGBoost model, and applied the SHAP value as a proxy to
quantify the relative importance of different FATES parameters.




**2.4 Overall experimental design**
Our experimental design flowchart is shown in Figure 2. Procedure "P1" in Fig. 2 is used to
generate an ensemble of parameter values for each experiment ensemble, i.e., Exp-1, Exp-2, and
Exp-3. First, a number of initial parameter sets (e.g., 5000 sets) were generated using Latin
Hypercube Sampling (LHS; Mckay et al., 2000). Second, the initial parameter sets were filtered
by the trait tradeoffs between early and late successional PFTs (Figure 1). We repeatedly increased
the number of initial parameter sets in the first step until 1500 parameter sets were obtained in the
second step. Each ELM-FATES experiment starts from bare ground and runs for 350 years to
reach an equilibrium state, by cycling the meteorological forcing during 2002–2005, and the last
four years of the simulations were analyzed.

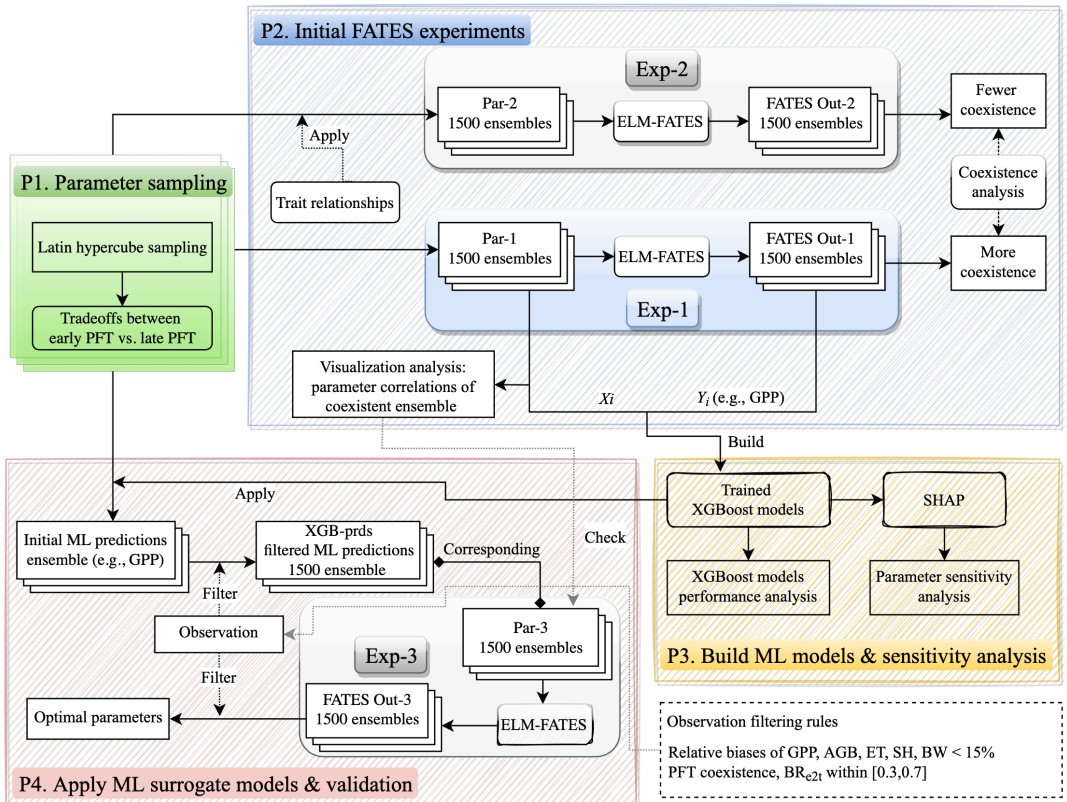




Figure 2. Overall flowchart of experimental design and associated analysis.
To test whether plant trait relationships established from field measurements can improve the
ELM-FATES simulations, two sets of experiment ensembles, i.e., Exp-1 and Exp-2 (procedure
"P2" in Figure 2), were conducted using two parameter ensembles (i.e., Par-1 and Par-2). For Par-
1, 1500 parameter sets were generated from procedure "P1" based on the entire eleven parameters'
space (i.e., $V_{cmax,early}$, $V_{cmax,late}$, $SLA_{early}$, $SLA_{late}$, $M_{bk,early}$, $M_{bk,late}$, $WD_{early}$, $WD_{late}$,
$L_{leaf,early}$, $L_{leaf,late}$, $CR_{s2l}$ ). For Par-2, 1500 parameter sets were generated from procedure "P1"
but only based on five parameters' space (i.e., $V_{cmax,early}$, $V_{cmax,late}$, $SLA_{early}$, $SLA_{late}$, $CR_{s2l}$).
The other six parameters ($M_{bk,early}$, $M_{bk,late}$, $WD_{early}$, $WD_{late}$, $L_{leaf,early}$, $L_{leaf,late}$,) in Par-2
were calculated based on the traits relationships defined by Equations (1) ~ (3). Therefore,
compared to Par-1, the parameters in Par-2 are constrained by the observed trait relationships. The
distributions of these two parameter sets are shown in Figure S1. $V_{cmax}$, $SLA$, and $CR_{s2l}$ have
similar distributions between Par-1 and Par-2. Compared with Par-1, Par-2 has a narrower
distribution of $M_{bk}$ but broader distributions of $WD$ and $L_{leaf}$.

Exp-1 and Exp-2 each include 1500 350-year ELM-FATES simulations. We averaged the last four
years of these simulations for analysis, i.e., outputs: Out-1 and Out-2, respectively. To quantify
the PFT coexistence, we computed the biomass ratio between early successional PFT and the total
biomass, denoted as $BR_{e2t}$. For brevity, **we denote the ELM-FATES experiments with $BR_{e2t} \in$**
**$[0.1, 0.9]$ as "coexistence", $BR_{e2t} \in [0.0, 0.1)$ as "late", $BR_{e2t} \in (0.9, 1.0]$ as "early"**. We
calculated $BR_{e2t}$ based on Out-1 and Out-2, and then computed the fraction of coexistence
experiments in each ensemble. As we will show in section 3.1, considering the observed trait
relationships, Exp-2 has a lower fraction of coexistence experiments. Therefore, only Exp-1 was





used for further ML-related analysis. We also performed some analysis of Exp-1 to explore
whether the parameters of the coexistence experiments have correlations with each other (Section

320    3.2).


Based on Exp-1, we trained XGBoost models to emulate the ELM-FATES model behavior and
analyzed the parameter sensitivity using SHAP (procedure "P3" in Figure 2). Sixteen variables
were used as XGBoost model features, including 11 parameters in Par-1 and 5 parameter
differences between early and late successional PFTs. The corresponding ELM-FATES annual
average outputs were used as XGBoost model targets. Specifically, six models were built, i.e.,
XGB_ET, XGB_SH, XGB_BW, XGB_GPP, XGB_AGB, XGB_BR for predicting ET, SH, BW,
GPP, AGB, and $BR_{e2t}$, respectively. Taking $BR_{e2t}$ as an example, the 1500 pairs of sixteen
features and the corresponding simulated $BR_{e2t}$ were randomly split into two groups, 90% used
for training and the remaining 10% used for testing. In the simulations of Exp-1, the coexistence
experiments only account for 20.6% (see Section 3.1 for details). Therefore, 90% of data is used
for training to ensure sufficient coexisting samples used in the training process. The choice of
hyperparameters in the XGBoost model can significantly impact its performance. In training, we
used the Bayesian optimization method to efficiently tune the XGBoost model (Snoek et al., 2012).
Additionally, a five-fold cross-validation method was utilized to avoid overfitting in the
hyperparameters optimization (Feigl et al., 2021), and the mean squared error was used as the
objective function. The root mean squared error (RMSE) and R-squared ($R^2$) are used to quantify
the overall XGBoost model performance for the training and testing data prediction. Furthermore,
based on the trained XGBoost models, we applied SHAP to identify feature importance to quantify
the parameter sensitivity of ELM-FATES.




The trained XGBoost models were then used to help select ELM-FATES parameters (procedure
"P4" in Figure 2). First, initial parameter sets were generated from procedure "P1" based on the
entire eleven parameters' space (Table 1, identical to the parameters' space used for the generation
of Par-1). Second, these parameter sets and parameter differences were sent to six XGBoost
surrogate models to predict ET, SH, BW, GPP, AGB, and $BR_{e2t}$. Third, the predictions were
further filtered by two criteria: (1) compared to observations, the relative biases of the predicted
ET, SH, BW, GPP, and AGB should be less than 15%; (2) the XGBoost model predicted $BR_{e2t}$
should be within [0.3, 0.7]. We repeated these three steps until we obtained 1500 sets of XGBoost
model predictions that match the criteria. Finally, we obtained 1500 sets of XGBoost model
predictions and their corresponding 1500 sets of parameters (Par-3). We also checked whether the
selected Par-3 can match the empirical relationships derived from the empirical analysis in
procedure "P2" (see Sections 3.2 and 3.5 for details). Then, the 1500 sets of parameters in Par-3
were sent to ELM-FATES to conduct 350-year runs (i.e., Exp-3). The last four years of the
simulations were averaged (i.e., Out-3) for further analysis. We then compared Out-3 with
observations and analyzed the PFT coexistence to obtain the optimal ELM-FATES parameters.






**3. Results**
**3.1 Comparison between Exp-1 and Exp-2**
Constraining the input traits using the observed trait relationships yields slightly better ELM-
FATES simulations of water, energy, and carbon variables (Figures 3a~3e). The distributions of
the relative biases of ET, SH, BW, and GPP have similar ranges between the two sets of
experiments (Figures 3a~3d). Compared with Exp-1, the 50[th] percentiles of relative biases of ET,
SH, BW and GPP for Exp-2 (with constrained traits) are closer to zero, indicating Exp-2 is slightly
better than Exp-1. The distribution of simulated AGB for Exp-2 is much narrower than Exp-1
(Figure 3e), which could be due to the narrower distribution of $M_{bk}$ (Figure S1).
Exp-1 has a much higher fraction of PFT coexisting simulations than Exp-2 (Figure 3f and Table
S2). Overall, 70.6 % of experiments in Exp-1, and 94.5% of experiments in EXP-2 have high
simulated $BR_{e2t}$ that is greater than 0.9. This indicates that both Par-1 and especially Par-2 favor
the early successional PFT. As for the coexisting experiments with $BR_{e2t} \in [0.1, 0.9]$, Exp-1 has
about five times more coexisting experiments (20.6%) than Exp-2 (4.1%). Further filtering the
coexisting cases by observations (Table S1), only 21 experiments remain in Exp-1, and 6
experiments in Exp-2 (Table S2). Even though Exp-2 considered the observed trait relationships,
it has fewer coexisting cases within the reasonable observation ranges than Exp-1. Therefore, Exp-
2 is not used in our remaining analysis.



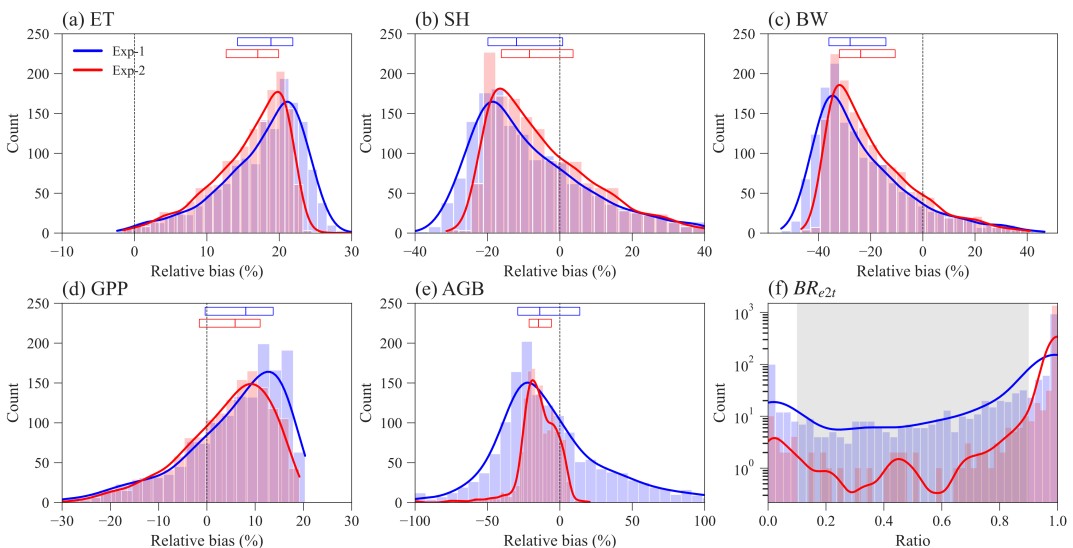

Figure 3. Distribution of ELM-FATES simulations for Exp-1 and Exp-2. The y-axis in (f) is

logarithmic. $Relative\ bias = \frac{simulation - observation}{observation} \times 100$ (%). In (a)~(e), the top horizontal

bars with three vertical lines denote the relative bias at the 25th, 50th, and 75th percentiles,

respectively. The grey shaded area in (f) represents the coexistence biomass ratio between 0.1

and 0.9.

**3.2 Parameter analysis of Exp-1**

We also tested whether simple parameter correlations can be constructed to guide the simulation

of PFTs coexistence. No simple parameter correlations can be built to distinguish the coexisting

cases from the early and late cases in Exp-1 (Figures 4, S2, and S3). Most parameter (or parameter

difference) spaces show large overlaps between early, late, and coexisting cases (Figures S2 and

S3). Notably, we empirically built three linear equations based on the boundaries in the parameter

spaces for the coexisting cases (Figure 4). Coexisting cases are primarily located in spaces with

$SLA_{late} > 0.35 \times SLA_{early} + 0.003$ (Figures 4a and 4d), $V_{cmax,diff} < -4800 \times SLA_{diff} +$

$100$ (Figures 4b and 4e), and $WD_{diff} > 55 \times SLA_{diff} - 1.3$ (Figures 4c and 4f), where



$V_{cmax,diff} = V_{cmax,early} - V_{cmax,late}$, and $SLA_{diff}$ and $WD_{diff}$ are defined likewise. Within
these constrained parameter spaces, the percentage of coexisting cases increases from the original
20.6% (i.e., 309 out of 1500) to 32.6% (i.e., 304 out of 932). Therefore, these empirical correlations
could help guide ELM-FATES parameter selection for coexisting PFTs. On the other hand, a
dominant proportion (i.e., 67.4% (1–32.6%)) of experiments are still either early or late cases
within the constrained parameter spaces and cannot robustly predict PFT coexistence. Moreover,
despite further considering the observational constraints (black scatters in Figure 4; Table S2), the
21 experiments (2.3%, 21 out of 932) are still sparsely distributed in the parameters' space of the
coexisting cases, so no simple correlations can be developed based on these simulations. Therefore,
simple empirically built relationships between plant traits provide limited benefit to guiding ELM-
FATES parameter selection for modeling PFTs coexistence while matching the observations. This
finding provides additional motivation for the ML-based approaches.

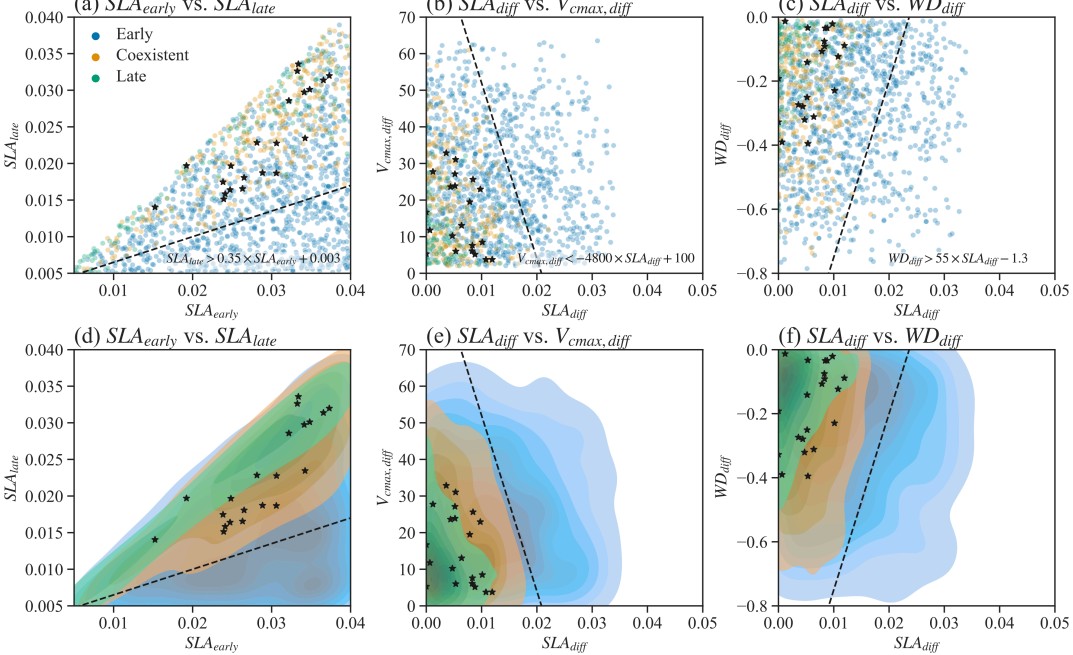






Figure 4. Relationships between selected parameters of Par-1. These parameters are presented in three groups, i.e., green color for the late cases with $BR_{e2t} \in [0.0, 0.1)$, orange color for the coexisting cases with $BR_{e2t} \in [0.1, 0.9]$, and blue color for the early cases with $BR_{e2t} \in (0.9, 1.0]$. Black star represents coexistence cases further filtered by observational constraints. (d)~(f) are the corresponding kernel density estimate plots of the scatter plots (a)~(c). $V_{cmax,diff} = V_{cmax,early} - V_{cmax,late}$. $SLA_{diff}$ and $WD_{diff}$ are defined likewise.

## 3.3 XGBoost model performance

Overall, the XGBoost surrogate models show good performance in predicting ELM-FATES simulations (Figure 5). Based on Exp-1 (i.e., Par-1 and Out-1), six XGBoost models were trained. In training, the RMSEs for the six models are zero or nearly zero, and $R^2$s are close to one. In the testing, four XGBoost models (i.e., XGB_ET, XGB_SH, XGB_BW, XGB_GPP) still show good performance with small RMSE and large $R^2$ (>0.95). XGB_AGB shows a little degradation with $R^2$ of 0.88. The performance of XGB_BR also shows degradation with $R^2$ decreasing from 1.0 in training to 0.75 in testing. XGB_BR cannot well predict the ELM-FATES simulated $BR_{e2t}$ of 0 or 1 when only one PFT survives. This indicates that PFT competition processes in ELM-FATES, which determine $BR_{e2t}$ and AGB, are highly nonlinear and difficult to emulate even using a state-of-the-art machine learning algorithm.





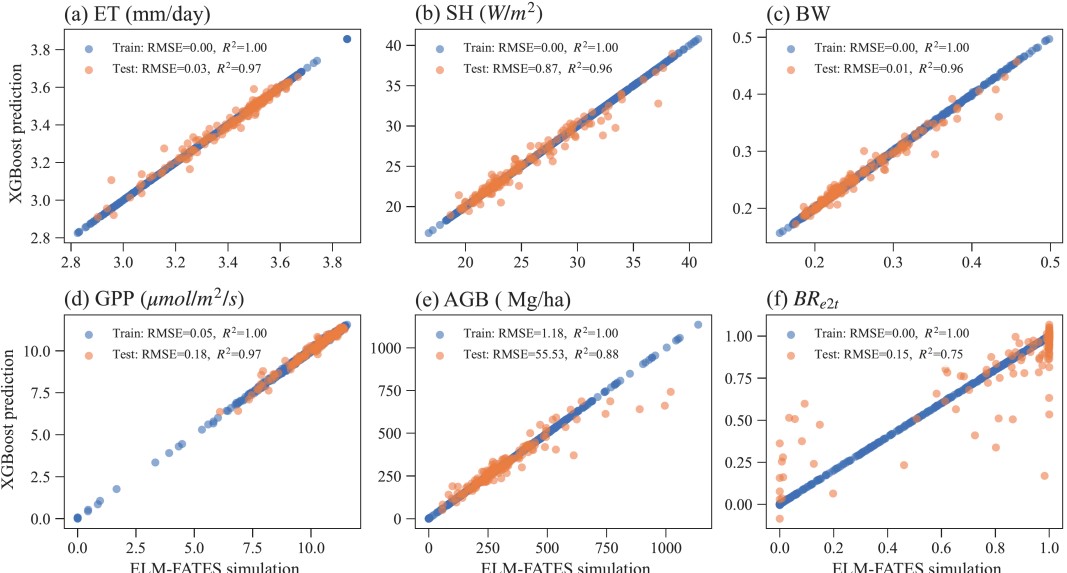


Figure 5. The performance of XGBoost surrogate models in the training and testing for
predicting (a) ET, (b) SH, (c) BW, (d) GPP, (e) AGB, and (f) $BR_{e2t}$.

**427 3.4 SHAP parameter importance analysis**

Figure 6 shows the feature importance, including parameters and parameter differences, for
different XGBoost models. Features (on the y-axis) with a higher mean absolute SHAP value (on
the x-axis) denote a larger contribution to the XGBoost model prediction. The number of most
important features is different for predicting ET, SH, BW, and GPP compared with predicting
AGB and $BR_{e2t}$.
For the XGBoost models that predict ET, SH, BW, and GPP, the top three features have the largest
SHAP values compared to the rest (Figures 6a~5d). Notably, these top three features are the same
and correspond to the early successional PFT, i.e., $V_{cmax,early}$, $SLA_{early}$, $L_{leaf,early}$. Most ELM-
FATES experiments in Exp-1 used as the training samples for the XGBoost models are early cases.
Therefore, the parameters of early successional PFT have dominant contributions in the XGBoost



model predictions of overall grid-level fluxes. These three parameters are positively correlated
with ET and GPP and negatively correlated with SH and BW (red vs. blue bars in Figures 6a~d;
Figure S4 for more details), reflecting the fundamental carbon metabolism of the typically
dominant early successional plant.
For the XGBoost surrogate models of AGB and $BR_{e2t}$, more than eight features have large SHAP
values (Figures 6e and 6f). Both early and late successional PFT parameters contribute to
predicting the two variables. Compared with the predictions of ET, SH, BW, and GPP with only
three major features, predicting AGB and $BR_{e2t}$ is relatively more complex. This is because AGB
and particularly $BR_{e2t}$ are closely related to the PFT competition process in which both the early
and late PFT traits are crucial. Especially for $BR_{e2t}$, the most important features are the parameter
difference between the early and late successional PFTs. For example, $SLA_{diff}$ is positively
correlated to $BR_{e2t}$. Therefore, to have coexisting PFTs with $BR_{e2t} \in [0.1, 0.9]$, the SLA of two
PFTs should neither be too large nor too small.




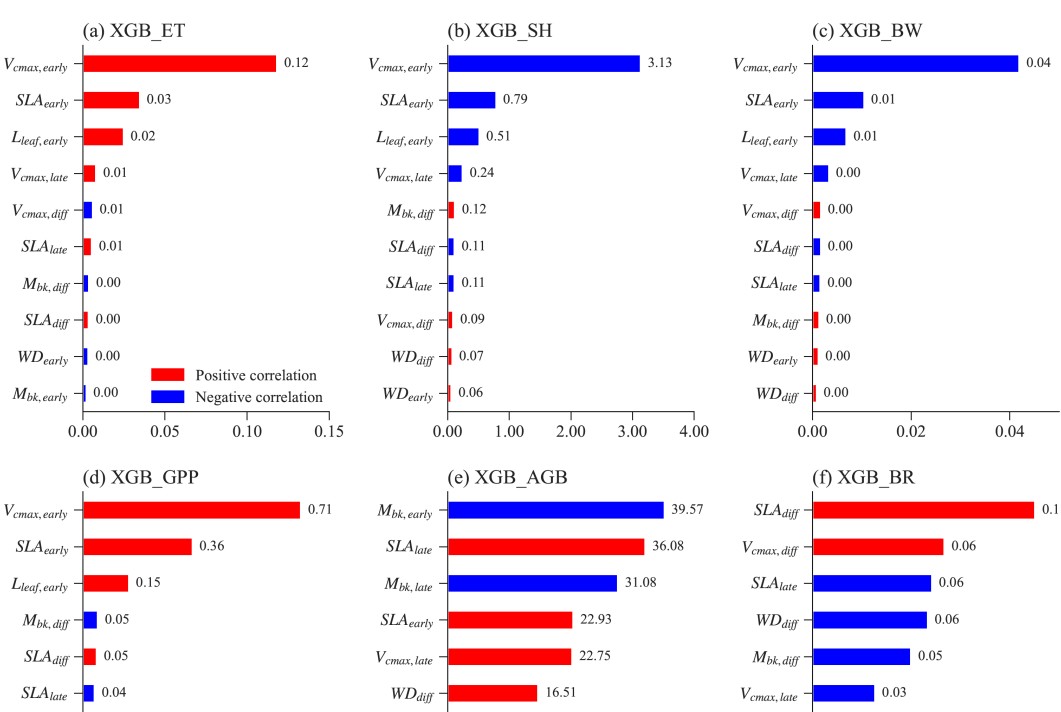

Figure 6. Mean absolute SHAP values for different XGBoost surrogate models for the top ten most important features. Absolute SHAP values are sorted in decreasing order from top to bottom. For each feature (y-axis) in each XGBoost model, the Spearman correlation coefficient is calculated between the feature values and the corresponding SHAP values (Figure S4). The red color means that a given feature is positively correlated with the predicting variable, whereas blue denotes a negative correlation.





**3.5 XGBoost model parameter selection**
Using the XGBoost surrogate models, the Par-3 ensemble was selected, including 1500 sets of
parameters and the corresponding parameter differences between the early and late successional
PFTs (Section 2.4, procedure "P4" in Figure 2). We examined whether Par-3 matches the empirical
relationships shown in Figure 4 (Section 3.2), i.e., $SLA_{late} > 0.35 \times SLA_{early} + 0.003$ ,
$V_{cmax,diff} < -4800 \times SLA_{diff} + 100$ , and $WD_{diff} > 55 \times SLA_{diff} - 1.3$ . In total, 99.1%
(1486 out of 1500) of parameter sets are consistent with the empirical relationships, indicating the
XGBoost models implicitly learned these simple relationships.
The parameter distributions of Par-3 show different patterns from the early/late parameters of Par-
1 (green vs. blue regions in Figure 7), but there are large overlaps between the coexistence
parameters of Par-1 and Par-3 (orange vs. green regions, e.g., the third column in Figure 7). This
indicates that the XGBoost surrogate models learned to select parameters around the parameters'
space of the coexisting cases. Par-3 also tends to have a smaller parameter difference between the
early and late successional PFTs in terms of $SLA_{diff}$ and $V_{cmax,diff}$. However, Par-3 also shows
different patterns from the coexisting parameters of Par-1, probably because the XGBoost selected
parameters were also constrained by multiple observations and implicitly considered parameter
tradeoffs. For example, the $V_{cmax,early}$ and $V_{cmax,late}$ of Par-3 are located in narrower ranges than
the coexisting parameters of Par-1 (first two columns in Figure 6).



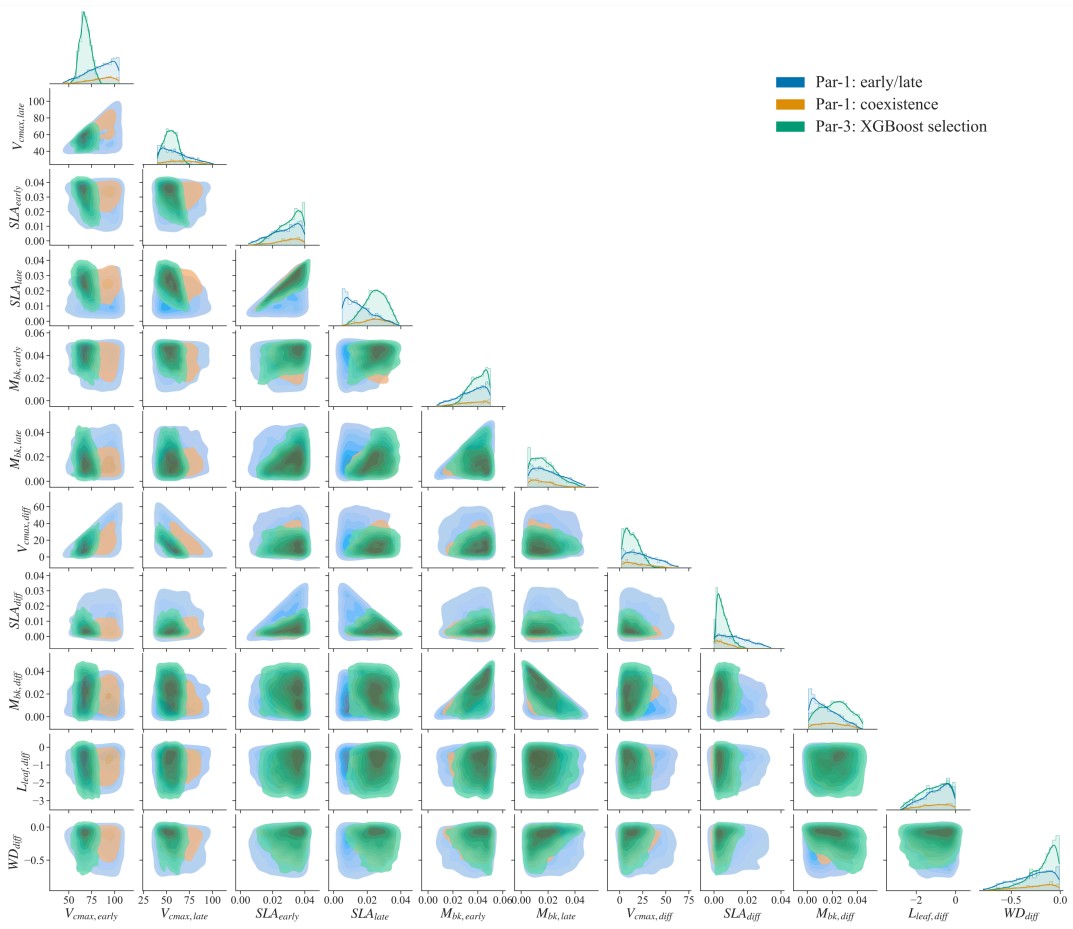

Figure 7. Comparison of parameter or parameter difference in Par-1 vs. Par-3 for eleven features. The diagonal plots represent each parameter's distribution, and the rest of the subplots are kernel density estimate plots. There are three groups, i.e., blue for the early/late cases of Par-1, orange for the coexisting cases of Par-1, and green for Par-3 selected by XGBoost models.





**3.6 Validation of ML selected parameters**
ELM-FATES simulations of Exp-3 based on the ensemble parameters of Par-3 selected by the
XGBoost surrogate models can better capture the observations and have more coexisting cases
than Exp-1 (Figure 8). The median values of simulated variables for Exp-3 are closer to
observations with relative biases closer to zero than Exp-1 (Figure 8a, blue vs. green boxes). The
Exp-3 simulated variables also have more concentrated distributions than Exp-1. Compared to the
skewed distribution of $BR_{e2t}$ in Exp-1 with a large proportion of early cases, Exp-3 has a more
normally distributed $BR_{e2t}$ (Figure 8b). Specifically, Exp-3 has about 3.6 times more coexisting
cases than Exp-1, i.e., 73.1% (1097 out of 1500) in Exp-3 vs. 20.6% (309 out of 1500) in Exp-1
(Table S3). After being further constrained by observation (Table S3), one-third of the experiments
(i.e., 495 out of 1500) in Exp-3 remain, and this ratio is 23.6 times more than 1.4% (21 out of 1500)
in Exp-1.
The XGBoost surrogate model predicted variables also match well with those simulated using
ELM-FATES in Exp-3 (Figure 8, orange vs. green boxes), indicating the overall reasonable
accuracy for the XGBoost model predictions. Compared to the ELM-FATES results using Par-3,
the XGBoost models show better performance for ET, SH, BW, and GPP, but relatively degraded
performance for AGB and $BR_{e2t}$ (Figure S5). It is consistent with the performance of the XGBoost
models' training and testing results (in Section 3.3).





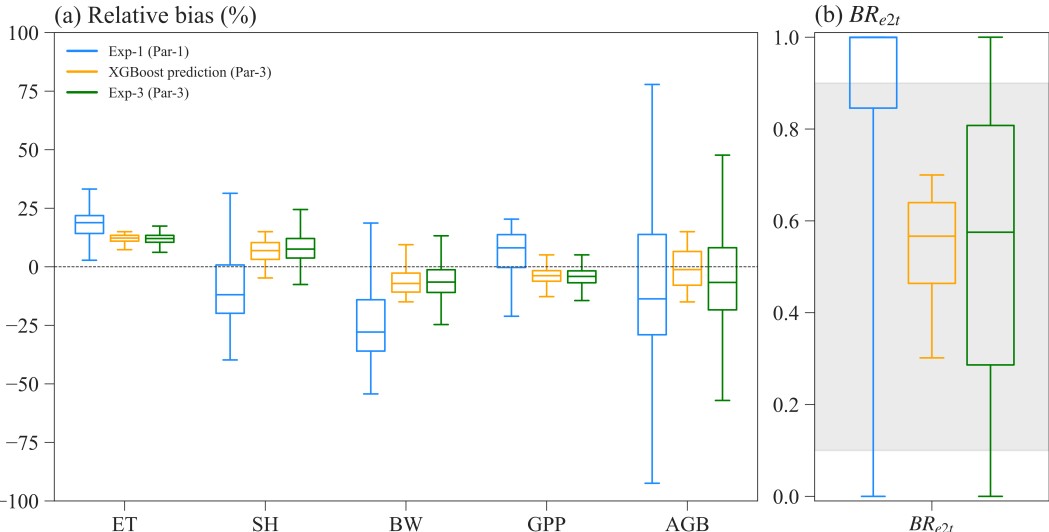

Figure 8. Comparison between the ELM-FATES simulations for Exp-1 and Exp-3. (a) Relative bias for simulated ET, SH, BW, GPP, and AGB. (b) Simulated $BR_{e2t}$. XGBoost prediction represents the selected XGBoost model predictions after filtering with observation and biomass ratio (i.e., the XGB_prds, procedure "P4" in Figure 2).

**3.7 Parameter tradeoff for coexisting experiments**

Parameters of the early and late successional PFTs show tradeoffs for the coexisting experiments. Large relative differences in $SLA$, $V_{cmax}$, and $WD$ (more negative) favor the early successional PFT, while large relative differences in $M_{bk}$ and $L_{leaf}$ favor the late successional PFT. Therefore, in Exp-1, compared to the early and late cases, the coexisting cases have intermediate relative differences in $SLA$, $V_{cmax}$, $WD$, $M_{bk}$ and $L_{leaf}$ (dashed boxes in Figure 9). The coexisting cases in Exp-3 have similar patterns with intermediate relative differences in $SLA$, $V_{cmax}$ and $L_{leaf}$ compared to the early and late cases (solid boxes in Figure 9). However, $M_{bk}$ and especially $WD$ show the largest relative difference for the coexisting cases compared to the early and late cases



in Exp-3. These two parameters still show a tradeoff in determining coexisting PFTs, because
larger $WD$ favors the early PFT while larger $M_{bk}$ favors the late PFT.

In Exp-3, the parameter spaces of the coexisting cases show large overlaps with the early/late cases
(Figure S6). There are no simple correlations between these parameters to distinguish the
coexisting cases from the early and late cases (also see Section 3.2). Although $WD_{diff}$ of the
coexisting cases still overlap with the early/late cases, when $WD_{diff}$ is less than roughly –0.4
(g/cm$^3$), only coexisting cases exist (Figure S6). Nevertheless, this rule (i.e., $WD_{diff} < -0.4$) alone
cannot ensure PFT coexistence (see Figure 7).

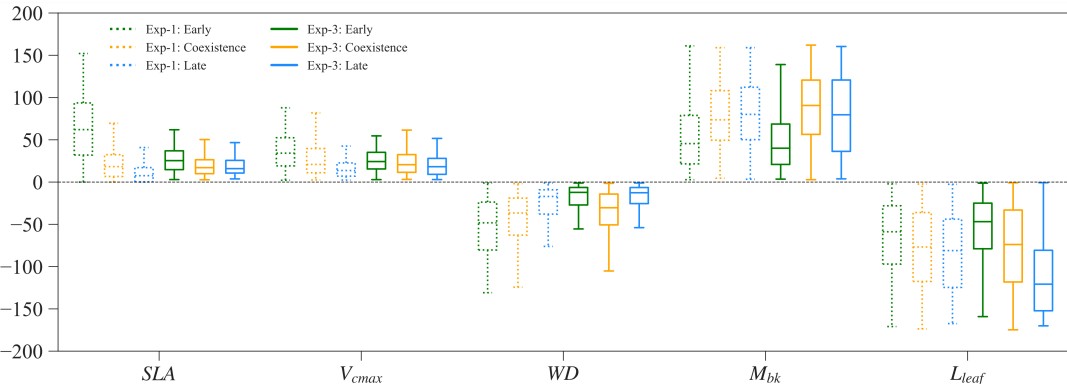

Figure 9. Parameter relative difference (%) between early successional PFT and late successional
PFT for Exp-1 (box with dash line) and Exp-3 (box with solid line). Parameter relative difference
is calculated as, taking SLA as an example, $\frac{SLA_{early} - SLA_{late}}{(SLA_{early} + SLA_{late})/2} \times 100$ (%).






**3.8 Seasonal variation comparison**

Figure 10 shows the seasonal variations of ET, SH, BW, and GPP for observations and simulations

of the finally selected 495 experiments in Exp-3 with good model performance (Table S3). Overall,

the simulated ET shows a similar seasonal variation to ET observation (Figure 10a), with relatively

small ET in the wet season (November–May), high ET in the dry season (June–October), and ET

peaks in August. However, compared to the observations, ELM-FATES overestimates ET,

especially during the wet season. The simulated SH also shows a similar seasonal variation with

the SH observation except in March. ELM-FATES overestimated SH from January to May but

underestimated SH from September to December (Figure 10b). Due to the discrepancy between

simulated ET and SH, the model underestimates BW from September to December (Figure 10c).

The simulated GPP has minor seasonal variability compared to the observed GPP. ELM-FATES

overestimates GPP from June–August in the dry season, but underestimates GPP over October–

December. The lower GPP over June–August indicates that plants may be relatively water-stressed

or energy limited during these months. However, the large ET observation over the same period

implies that this site is unlikely water limited or strongly energy limited. The ELM-FATES

simulations also display little water stress year-round (Figure S7). Therefore, there are likely

elements of the seasonal cycle (e.g., phenological responses of photosynthetic capacity) that are

not yet captured here. Additionally, tower estimates of GPP may also have large uncertainties.



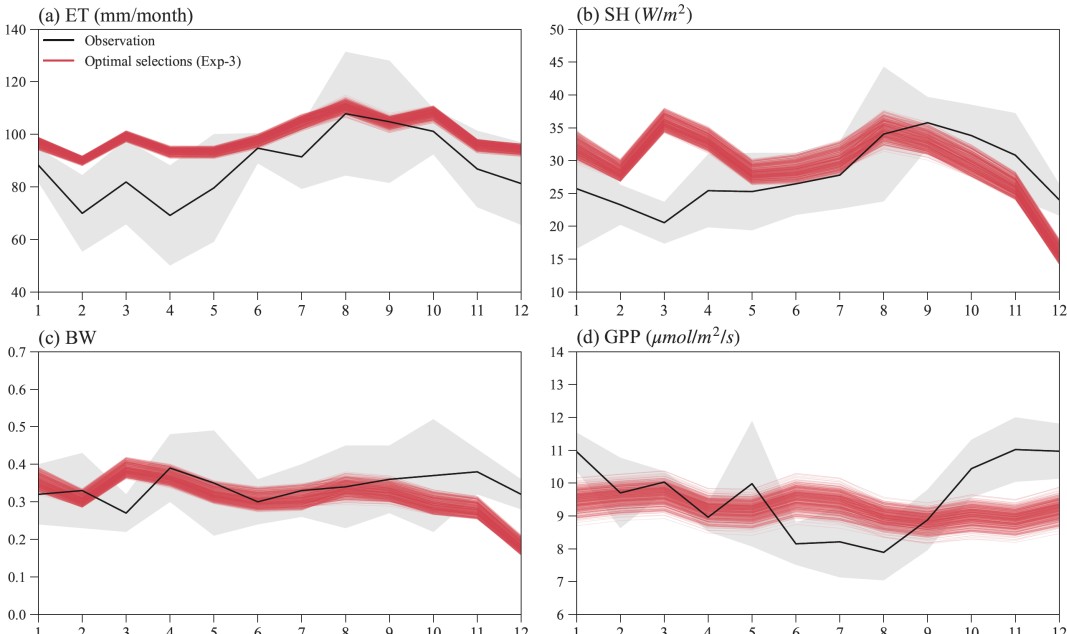

Figure 10. Mean monthly observations and selected optimal ELM-FATES simulations in Exp-3

for (a) ET, (b) SH, (c) BW, and (d) GPP. Each red line represents one experiment simulation (four-

year simulation average). The black curves are monthly climatologic averages from 2000 to 2008,

and the grey shaded area represents the interannual variabilities (i.e., $mean \pm$

$standard\ devation$).





## 4. Discussion

### 4.1 Limited guidance of observed trait relationships for PFT coexistence modeling in FATES

We found degraded PFT coexistence in ELM-FATES simulation when observed trait relationships are considered. More specifically, constrained by observed trait relationships, Exp-2 has fewer coexisting cases than Exp-1 which does not consider the observed trait relationships. The observed trait relationships were derived from site measurements in the species-rich tropical ecosystem where plant coexistence commonly happens (Kraft et al., 2008), which is expected to enhance the PFT coexistence simulations. This inconsistency could be due to several possible reasons. First, ELM-FATES is a typical "trait filtering" model (Fisher et al., 2018), and the realistic simulation of PFT dynamics largely depends on the fidelity with which trait tradeoff surfaces are prescribed in the model (Scheiter et al., 2012). Implicit representation of trait tradeoff in the current ELM-FATES model may not be well balanced, which may differ from the observed trait relationships that lead to coexistence in the real world (at least for the ecosystem at our study site). In particular, there may be correlated tradeoffs that are unmeasured (e.g., with below ground processes, Chitra-Tarak et al. 2021) but not represented in the model. A second reason could be the mismatch between different spatial scales. The observed trait relationships are derived from field measurements across tropical forests over a large region with diverse species and climate, e.g., the relationship in equation (1) is for plant species in Panama. In contrast, ELM-FATES simulations were conducted at the K34 site scale with specific species composition. Therefore, the large-scale trait relationships may not reflect the small-scale trait relationships. Wright et al. (2005) showed that trait relationships fitted for individual sites varied considerably. Third, the observed trait relationships are based on simplified equations, which may not be able to comprehensively reflect PFT coexistence. For example, although equation (2) derived from Longo et al. (2020) can reflect





the negative relationship between SLA and $L_{leaf}$, the $R^2$ of this equation is about 0.49, which may
not be accurate enough to represent trait relationships. Additionally, these equations (1)~(3) do not
consider the uncertainty of traits covariance. In Koven et al. (2020), the uncertainties between trait
covariance were considered when sampling parameters for FATES experiments, which may be
considered in future studies.

**4.2 Advantages of ML surrogate models on improving PFT coexistence modeling**
ELM-FATES simulations driven by parameters selected using the XGBoost models essentially
improved PFT coexistence and better captured observations. Compared to the initial Exp-1, which
was used to train the XGBoost models, the proportion of coexisting PFTs in Exp-3 reaches 73.1%,
3.6 times more than 20.6% in Exp-1. Further filtering the coexistence experiments by observations,
Exp-3 still has 33.0% of experiments left with good model performance, 23.6 times that of 1.4%
of experiments in Exp-1 with good performance. Our ML-based approach also outperforms the
empirical correlations built in Section 3.2, which only yields 32.5% of coexistence experiments
and this reduces to 2.3% of experiments if further constrained by observation. The large proportion
of optimal experiments selected by our ML approach also outperforms previous studies using
direct filtering approaches. Buotte et al. (2021) conducted two stages of experiments to select
optimal parameters for CLM-FATES modeling with two conifer species; only 0.3% (1 out of 360)
of the cases met the given criteria in the first stage experiments, which increased to 5.5% in the
second stage experiments. Huang et al. (2020) conducted CLM-FATES modeling with two
tropical PFTs at the Tapajós National Forest sites; only one parameter set out of seventy (about
1.4%) was selected with reasonable fractions of two PFTs and minor errors compared to
observations. In addition, the parameter selection procedures of these two studies require some



degree of subjective decision making and expert knowledge. On the other hand, our ML-based
approach takes a more objective procedure, and little expert knowledge is required except for the
initial determination of the parameter reference ranges. Importantly, we believe this approach can
be repeatable as, e.g., model developments lead to changes between the parameter values and
model predictions of forest structure and function, and can be used to define constrained ensemble
values that will allow assessment of confidence in model predictions. Even though simulating
coexistence of different plants may not be a big concern for individual-based DGVMs , e.g.,
LPJmL-FIT (Sakschewski et al., 2015, 2016) and TROLL (Maréchaux and Chave, 2017), our
approach also could be applied to the selection of key parameters that regulate vegetation dynamics
in these models.

Our study also reproduced the observations satisfactorily. Holm et al. (2020) conducted the ELM-
FATES simulation with only one PFT considered at the same K34 site. Our study yields better or
similar performance in the magnitude of AGB, and the magnitude and seasonal variation of GPP,
ET, SH, and BW (Table 2 and Figure 3 in Holm et al. 2020 vs. Figures 8 and 10 in this study). It
should also be noted that the overestimation of simulated energy fluxes (latent heat and SH) from
January to May could be associated with the energy-related processes (e.g., energy partition,
surface albedo) in ELM-FATES. Other potential reasons could be related to the uncertainties in
atmospheric forcing and the common issue of incomplete energy budget closure at eddy covariance
towers (Wilson et al., 2002; Foken, 2008; Rocha et al., 2009).

Compared to the predictions of GPP, ET, SH, and BW simulated by ELM-FATES, the XGBoost
surrogate models show slightly degraded performance in predicting the simulated $BR_{e2t}$ and AGB



(Figures 5 and S5). Three parameters ($V_{cmax,early}$, $SLA_{early}$, and $L_{leaf,early}$) mainly control the
predictions of ET, SH, BW, and GPP, while eight features are crucial for predicting AGB and
$BR_{e2t}$. Even though the XGBoost algorithm has an excellent ability to capture complex nonlinear
relationships, it does not predict well the PFT competition related variables of AGB and $BR_{e2t}$
because the physical model cannot robustly predict coexisting PFTs due to the higher
dimensionality of predicting PFT composition as compared to other ecosystem variables.
Therefore, even though the XGBoost surrogate models essentially improve plant coexistence
modeling, further studies are still needed to improve the emulation of PFT competition related
variables. Other approaches that have been applied in DGVMs but not specifically for PFT
coexistence modeling, for example, the generalized likelihood uncertainty estimation (GLUE)
approach (Zhang et al., 2022) and the Bayesian model emulation approach (Fer et al., 2018), could
provide alternative ways. Additionally, the adoption of deep learning algorithms and the
consideration of additional mechanisms in FATES are also advocated.

**4.3 Trait tradeoffs between coexisting PFTs**
Trait-related parameters show tradeoffs between early and late successional PFTs for the ELM–
FATES simulated coexisting experiments. The relative differences between the two PFTs in $SLA$,
$V_{cmax}$, and $WD$ complementarily coordinate with the relative difference in $M_{bk}$ and $L_{leaf}$, hence
avoiding competitive exclusion (Figure 9). These ELM-FATES reflected tradeoffs are consistent
with the niche-based species coexistence mechanisms of environmental filtering and niche
partitioning (MICHALKO and PEKÁR, 2015; Adler et al., 2013). On the one hand, in the
coexisting cases, the relative differences between the two PFTs' parameters should not be
considerable. For example, a large difference in SLA more likely favors the early cases (green





dash box in Figure 9). This is related to environmental filtering in which coexisting species require
some degree of convergence in strategy to survive and persist under given environmental
conditions (Cadotte and Tucker, 2017; Thakur and Wright, 2017). On the other hand, some degree
of differences should exist between the two PFTs' parameters in the coexisting cases. This is
related to niche partitioning to ensure either difference in resource requirements or differences in
tolerance to surrounding conditions (Kraft et al., 2015; Fowler et al., 2013). Phenomenological
evidence has shown that functional trait variation promotes coexistence or increases species
richness (Uriarte et al., 2010; Angert et al., 2009; Adler et al., 2006; Mason et al., 2012; Ben-Hur
et al., 2012).

In our ELM-FATES simulations, the primary axis of competition for resources is light. The
tradeoffs between the two PFTs' parameters differentiate their vertical competition in light
absorption, which has been shown to strongly control tropical forest community composition
(Farrior et al., 2016; Poorter et al., 2003). Even though the early PFT has a shallower rooting depth
than the late PFT, there is no critical dry condition during our simulation period (i.e., corresponding
to values of the water stress factor (BTRAN) close to 1.0 in Figure S7). Therefore, competition for
water resource access negligibly contributes to PFT coexistence in this study. Previous tropical
studies also revealed these coexistence mechanisms. At a tropical forest site in eastern Ecuador,
Kraft et al. (2008) found that cooccurring trees are often less ecologically similar, and both
environmental filtering (different topographic habitats of ridgetops vs. valley) and niche
differentiation simultaneously contribute to species coexistence. Swenson & Enquist (2009) also
found that at small spatial scales in a tropical forest, most traits of coexisting species were under-





dispersed, consistent with environmental filtering, while the seed mass and maximum height were
over-dispersed, reflecting niche partitioning.

**4.4 Limitations and further model development**
Some limitations exist in our experiments. Niche partitioning is a critical aspect of promoting
species coexistence, which is closely related to spatial heterogeneity, temporal heterogeneity,
disturbances (e.g., nature enemy, fire), and resource partitioning (Adler et al., 2013). In our current
ELM-FATES simulations, some processes that have been or are being developed in the model are
not considered. These processes include nutrient limitation (Holm et al., 2020), fire disturbance
(Fisher et al., 2015), subsurface lateral flow (Fang et al., 2022), and plant hydraulics (Chitra-Tarak
et al., 2021; Li et al., 2021). Ignoring these processes could limit the potential of niche partitioning
among PFT in our ELM-FATES simulations. Topography has been recognized as an essential
spatial heterogeneity factor for tropical forests, but it is not considered in ELM-FATES (Kraft et
al., 2008; Costa et al., 2022). For example, Fang et al. (2022) coupled a three-dimensional
hydrology model (ParFlow) with ELM-FATES and found that lateral flow plays a prominent role
in governing aboveground biomass, and Cheng et al. (2021) also found a critical role for subsurface
hydrology on coexistence. As these processes are added to the model, the reproducibility aspects
of the XGBoost method to identify PFT combinations that match a broad range of criteria will be
particularly important.
Lacking other features or processes could also affect PFTs coexistence in the current FATES. For
example, plant trait plasticity, that plants can adjust their morphological and/or physiological traits
to better adapt to the environment (Nicotra et al., 2010; Bloomfield et al., 2018; McDowell et al.,
2022), is also not well considered in FATES. Leaf traits such as $V_{cmax}$ and SLA do vary vertically



through the canopy in FATES, via a prescribed relationship described by Lloyd et al., 2010. Liu
and Ng (2019) found that the SLA of a desert shrubland is significantly correlated with seasonal
water availability. Additionally, FATES only considers the inter-PFT variance of functional traits
(e.g., different $V_{cmax}$ for early and late PFT). However, studies revealed that trait variations
commonly exist within and between species (Wright et al., 2005; Engemann et al., 2016; Meng et
al., 2015; Dong et al., 2020; Siefert et al., 2015), which play a vital role in maintaining plant
diversity (Violle et al., 2012; Lu et al., 2017). Reproductive features that enhance competitive
exclusion tendencies have been illustrated to affect coexistence (Maréchaux and Chave, 2017;
Fisher et al., 2018). Hanbury-Brown et al. (2022) discussed the importance of the representation
of forest regeneration, including improving parameters and algorithms for reproductive allocation,
dispersal, seed survival and germination, environmental filtering in the seedling layer, and tree
regeneration strategies adapted to wind, fire, and anthropogenic disturbance regimes. Besides, both
growth-survival and stature-recruitment tradeoffs are critical to accurately predict successional
patterns in tropical forest structure and competition (see details in Rüger et al., 2020), which should
also be better considered in future model development. Furthermore, measured plant traits are
increasingly available, e.g., the TRY datasets (Kattge et al., 2020) can be used to improve the
model process and parameterizations. Future studies on properly and adequately using these
datasets to guide DGVMs parameterizations are advocated.



**5. Conclusions**
In this study, we explored two possible solutions to improve PFT coexistence modeling in a cohort-
based model (ELM-FATES): (1) using plant trait relationships established from field
measurements and (2) using machine learning based surrogate models to optimize parameters.
Multiple ensembles of ELM-FATES experiments were conducted over a tropical forest site at
Manaus, Brazil. We found that considering the observed trait relationships (Exp-2) slightly
improves the simulations of water (ET), energy (SH and BW), and carbon (GPP, AGB) variables
when compared against observations, but degrades the simulation of PFT coexistence. Based on
Exp-1, the XGBoost surrogate models were built to optimize the ELM-FATES parameters by
integrating the observations (i.e., ET, SH, BW, GPP, and AGB) and PFT coexistence criteria (i.e.,
PFT biomass ratio). Exp-3 with parameters selected by the ML-surrogate models vastly improves
the ELM-FATES simulation of PFT coexistence, and also better reproduces the annual means and
seasonal variations of ET, SH, BW, GPP, and the filed inventory of AGB. This study demonstrates
the benefits of using machine learning models to improve the modeling of PFT coexistence in
ELM-FATES and modeling of tropical forest environments, with important implications for
modeling the response and feedback of ecosystem dynamics to climate change. Our results also
suggest that adding additional mechanisms of species competition in FATES is also critical for
robust modeling of coexisting PFTs.



*Code and Data Availability.* The ELM-FATES source code, related surface and domain data, and
forcing data used in this study are archived on Zenodo (Li et al., 2022,
https://doi.org/10.5281/zenodo.7319876 ). The observational reference datasets of GPP, ET, SH,
BW, and AGB are obtained from Holm et al. (2020). The forcing data is available from Oak
Ridge National Laboratory Distributed Active Archive Center (ORNL DAAC), LBA-ECO CD-
32 Flux Tower Network Data Compilation, Brazilian Amazon: 1999-2006, V2,
https://daac.ornl.gov/LBA/guides/CD32_Fluxes_Brazil.html.

*Author contributions*. LL and YF designed and conducted the experiments, analyzed model
outputs, and drafted the manuscript. ZZ and MS contributed to the machine learning, experiment
design, and improvement of the manuscript. LRL contributed to the interpretation and discussion
of results, and improvement of the manuscript. ML, CDK, JAH, RAF, NGM, and JC contributed
to the dataset, interpretation and discussion of the results, and modification of the manuscript.

*Acknowledgments*. This research was conducted at Pacific Northwest National Laboratory,
operated for the U.S. Department of Energy by Battelle Memorial Institute under contract DE-
AC05-76RL01830. This study was supported by the Department of Energy's (DOE) Office of
Biological and Environmental Research as part of the Terrestrial Ecosystem Science program
through the Next-Generation Ecosystem Experiments (NGEE)-Tropics project.

*Financial support*. This research was supported by the U.S. Department of Energy, Office of
Science (grant no. 71073).
*Competing interests*. The authors declare that they have no conflict of interest.



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
