# Peer review of "A machine learning approach targeting parameter estimation for plant"

_EGUsphere, 2022_

## Author Response (AR1)

**Responses to comments <egusphere-2022-1286>**

Dear Editor and referees,

We greatly appreciate your insightful comments and suggestions, which have helped us improve our manuscript. In response to the referees' comments, we have made several significant changes in the revised manuscript, which are summarized below:

- We have updated the "Code and Data Availability" section and made the associated code and data available on Zenodo with a DOI.
- We have reorganized Sections 2.3 and 2.4, and added a new Section 2.5.
- We have added a new Section 4.5, summarizing the potential benefits of machine learning for VDMs modeling.
- We have revised the ensemble names of parameters, outputs, and experiments to make them more intuitive, and updated them throughout the manuscript.

The point-by-point responses to specific comments are provided below in blue. All line numbers listed below correspond to those in the clean version of the revised manuscript. We hope that our modifications have addressed all the concerns raised, and we appreciate your consideration of our revised manuscript.

Sincerely,

Lingcheng Li and co-authors

***Editor***

The Editor requested us to provide details on a suitable repository to deposit the relevant code and data and to include this information in the revised manuscript.

Thank you for the direction. We uploaded our relevant code and data to the Zenodo repository. Details for the repository are now included in the "Code and Data Availability" section of the revised manuscript now reads:

"Code and Data Availability. The ELM-FATES source code, surface and domain data, forcing data, and ML codes used in this study are archived on Zenodo (https://doi.org/10.5281/zenodo.7730685). The observational reference datasets of GPP, ET, SH, BW, and AGB are obtained from Holm et al. (2020). The forcing data are available from Oak Ridge National Laboratory Distributed Active Archive Center (ORNL DAAC), LBA-ECO CD-32 Flux Tower Network Data Compilation, Brazilian Amazon: 1999-2006, V2, https://daac.ornl.gov/LBA/guides/CD32_Fluxes_Brazil.html."

*Referee 1*

In this paper, the authors employ a machine-learning approach to optimize parameters from a vegetation demography model - FATES. Their approach clearly shows the bright application of ML as a tool to improve the next-generation Earth system models. The paper is very well written and the question being addressed is novel. I really enjoyed reading the manuscript and learned a lot from the authors. I would recommend accepting this paper in its current form.

Thank you for your positive feedback and for suggesting acceptance of our submission for publication.

*Referee 2*

This is an interesting study making use of machine learning to improve modelling parametrization in a complicated set-up of tropical forests with competing PFTs adopting different strategies. The manuscripts reads well and is clear and exhaustive. The paper is well suited for GMD and should be published. But before, I have the following suggestions to make, hopefully to improve it ...

Thank you for your encouraging comments and suggestions for improving our manuscript. We provided a response for each comment as detailed below.

L242: L_leaf is not defined yet in the text, and neither is WD.

Thanks for pointing these out. Their definitions are presented in Table 1. We have also added a clarification in L297, stating "(see Table 1 for parameter definitions)".

L260: perhaps explain what the "ensemble" is before. Help readers understand what makes XGBoost different from a standard Random Forest

We added an explanation of "ensemble" and briefly introduced the difference between XGBoost and RF:

In L253–L259. "Ensemble learning techniques combine the predictions of multiple independent base models (e.g., decision trees) to produce more accurate predictions, with popular algorithms such as Random Forest (Breiman, 2001) and XGBoost. While Random Forest builds an ensemble of parallel trees using bagging and produces the final prediction by averaging the outputs of all the individual trees, XGBoost sequentially trains a set of decision trees using boosting (Friedman, 2001), where each successive tree corrects the mistakes of its predecessors, and the final prediction is obtained by combining the predictions of all the trees using a weighted sum."

The following is a bit subjective, but I find the structure of the Methodology section slightly sub-optimal. I would have prefered to have an overall experimental design at the beginning, e.g. before talking about the SHAP values. This could be just an overview, not necessarily with all the details in section 2.4, but rather a general understanding of what will come up after. Furthermore, section 2.4 is very long compared to the other ones, so splitting it or rebalancing them would be good.

Thanks for the suggestions.

After careful consideration, we think Section 2.3 Machine Learning Algorithms should be kept as an individual section, but the SHAP analysis can be moved to a new Section 2.5. We have also updated the structure of Section 2.4 Experiment Design by providing a short overview and subtitles to better organize the section. We added a new Section 2.5, which outlines the details of the ML model training and testing and includes the SHAP analysis removed from Section 2.3. Please refer to the changes in Section 2.3 in L249, Section 2.4 in L267, and Section 2.5 in L353.

In addition, to make the names of parameter and experiment ensembles more intuitive and understandable, we have replaced their names throughout the manuscript including the tables/plots. Xxx-CTR: refers to the control experiment ensemble WITHOUT considering the observed trait relationships.
Xxx-OBS: refers to the experiment ensemble WITH consideration of the observed trait relationships.
Xxx-ML: refers to the experiment ensemble WITH simulations guided by ML, using parameter values that were selected by the ML surrogate models.
Specifically,

| Old names | → | New names |
|---|---|---|
| "Exp-1", "Par-1", "Out-1" | | "Exp-CTR", "Par-CTR", "Out-CTR" |
| "Exp-2", "Par-2", "Out-2" | | "Exp-OBS", "Par-OBS", "Out-OBS" |
| "Exp-3", "Par-3", "Out-3" | | "Exp-ML", "Par-ML", "Out-ML" |

Also, when you present the observational relaitonships (Eqn 1 to 3) it does not seem entirely clear why this is done. While after the difference between Exp 1 and Exp 2 is mentioned, that is clear. I this it would read better if these observatinal relationships are introduced when you explain the experimental design (around line 300).

The description of the observational trait relationships has been moved to Section 2.4.2, spanning from L290 to L300.

L313: why is this in bold?

The bold formatting is removed.

L348: why 15%?

In Table S1 (and Figure 10), the annual means and their respective standard deviations of observed ET, GPP, SH, and BW were presented. The coefficient of variation (CV), which is the ratio between the standard deviation and mean, for ET, GPP, SH, and BW is 17.1%, 9.2%, 19.3%, and 26.4%, respectively. The CVs indicate the variability of observations, with an average of approximately 18.0%. Based on the variability of observations, we selected a relative bias threshold of 15% to filter the parameter selections.

L349: why [0.3, 0.7]?

Thanks for the question. The XGB_BR, the ML surrogate model for predicting $BR_{e2t}$, did not perform as well as the ML surrogate models for the other variables. We chose the range [0.3, 0.7] where XGB-BR has a better performance (smaller bias compared to the test dataset) (Figure 5). By doing so, we hope the ML surrogate model selected parameters can yield more ELM-FATES coexistence experiments.

In L333, "which corresponds to the range where the XGB-BR model exhibited relatively better performance (Figure 5)"

L374: What does this incapacity of Exp2 (with the observational constraints) to reproduce realistic coexisting ratios tell us about the standard model itself (ELM-FATES without the ML)? Does it reveal some underlying biases/problems in the structure of ELM-FATES itself that makes it less adapted to this tropical context for some reason? It would nice to get some insights on this to guide model development.

Thanks for the question and suggestion. As discussed in Section 4.1 (L574) of the original manuscript, we provided several explanations for why Exp-2 (now Exp-OBS) showed worse coexisting predictions. Additionally, in Section 4.4 (L703) of the original manuscript, we proposed potential avenues for further model development.

Following the referee's suggestions to incorporate insights for guiding model development, we added the following text to Section 4.1 (L600): "Furthermore, machine learning models can also be employed to extract the relationships between plant traits, which can then be incorporated into ELM-FATES and evaluated in future studies."

L422: The main problem here is probably the paucity of ELM-FATES simulations that are available to train the XGBoost, rather than the ML algorithm itself. To get highly non-linear behaviour, these require many more training samples. These would typically be trainend with many 1000s of simulations. You could mention that other ML techniques work better for sparser data (even when highly non-linear). I believe Gaussian Processes would be one good candidate to explore. (I am not saying this is needed in this paper, but it should be discussed + acknowledged).

Thanks for your suggestions. We added some discussions in Section 4.2.

L649. "Another important point worth mentioning is the small sample size of coexistence cases in Exp-CTR, with only 309 cases having $BR_{e2t}$ in the range of [0.1, 0.9], while the majority of cases

are dominated by either early or late successional PFT. This limited sample size may not provide enough data to train the XGBoost surrogate model sufficiently for predicting $BR_{e2t}$ within the range of [0.1, 0.9]. Therefore, further studies are still needed to improve the emulation of PFT competition related variables"

In L657, "Furthermore, we suggest exploring other machine learning algorithms, such as Gaussian process and neural network algorithms, which may be better suited for capturing non-linear correlations and learning from sparse data, and incorporating additional mechanisms in FATES to address some structural limitations."

L477: Why not place the orange coexistence above the green XGBoost, to increase clarity? or perhaps try some transparnecy?

Thanks! We increased the transparency to make it easier to distinguish between the different categories.

L525: Fig 9: There is something I am not following anymore... I must be missing something... why are there are both "early" and "late" boxes when the measure/index is precisely a difference between late and early? Not sure what I need to take home as a message from this plot.

In Fig. 9, "early/late" in the legend refers to the different experiment categories. We denote the ELM-FATES experiments with $BR_{e2t} \in [0.1, 0.9]$ as "coexistence", $BR_{e2t} \in [0.0, 0.1)$ as "late", $BR_{e2t} \in (0.9, 1.0]$ as "early". Each experiment includes parameters for early successional PFT and late successional PFT, which are used to calculate the parameter relative difference displayed on the Y-axis.

The purpose of the plot is to illustrate how parameter tradeoffs between early and late successional PFTs influence coexistence modeling. We have also discussed this in Section 4.3, where we highlight how the parameter tradeoffs align with the niche-based coexistence theory.

L549: Fig 10: Perhaps it would be wise to also add a bundle for the "standard" ELM-FATES without ML (ie. Exp 1? or also 2?) just to illustrate the improvements that the study proposes brings things closer to the observations.

For clarity, we have focused on showcasing the optimal cases selected from Exp-ML to demonstrate good performance in seasonal variations. While Exp-CTR (formerly Exp-1) is relevant, it only has 21 optimal cases, which is a lot fewer than the optimal cases in Exp-ML. Additionally, the difference between optimal cases in Exp-CTR and Exp-ML is primarily in terms of annual mean, rather than seasonal variation. We have already presented the annual mean analysis (i.e., relative bias) between Exp-CTR and Exp-ML in Table S3. Therefore, we have decided not to include Exp-CTR in Fig. 10.

The discussion is quite nice and exhaustive. However, it stays very focused on this modelling experiment and context. This is nice, but it would be (in my opinion) better to also have a more general overview going beyond this specific case of ELM-FATES on tropical forests of Manaus. To increase the breadth of the paper, I think a more general discussion on how to incorporate ML with process-based models would be very welcome. Inspirations could come from the perspective paper of Reichstein et al (2019) [https://doi.org/10.1038/s41586-019-0912-1]. For instance, it would be interesting to know if here the present study could fit in this logic of "hybrid-modelling" . There are also there other strategies to combine ML with ELM-FATES (as discsussed in the perpective paper) that could be evoked to outline further perspectives of this GMD manuscript. Finally, I would also encourage some more words of how much the auhtors beleive their approach is transferable to other contexts beyond tropical forests.

Thanks for your suggestions and the reference paper. We added one paragraph describing the general perspective on how machine learning can improve VDM modeling.

In L743:

"4.5 Enhancing VDM prediction with machine learning

We provide a brief overview of how machine learning can be applied to improve the modeling of plant dynamics, specifically in the context of vegetation demographic models. Firstly, ML can be used to derive trait parameter values. For instance, in this study, ML could be applied to replace the simple equations to derive the relationships between measured traits (Section 4.1). By integrating multiple datasets, including in situ measurements, atmospheric forcing, and remote sensing, ML could derive the spatial patterns and temporal variations of trait parameters for use in large-scale VDM modeling. Secondly, ML can be utilized to optimize parameters by developing surrogate models that emulate the relationships between the parameters and the VDM simulations, and using the surrogate models to identify optimal parameter values. This application has demonstrated success in this study and previous studies (e.g., Tsai et al., 2021; Dagon et al., 2020; Watson-Parris et al., 2021). Another benefit of using ML in VDMs is the ability to develop benchmark datasets. For example, studies have successfully employed ML to derive AGB datasets for various ecosystems (Morais et al., 2021; Zhang et al., 2020; Li et al., 2020; Bispo et al., 2020; Pham et al., 2020). These datasets can serve as benchmarks to evaluate the accuracy of VDM simulations. Lastly, ML can be used to replace semiempirical sub-models with little theoretical bases in DGVMs (Reichstein et al., 2019). For example, accurately modeling wildfire using process-based wildfire models integrated in DGVMs remains challenging. However, ML-based wildfire models have shown advantages in accuracy and computational efficiency (Rodrigues and Riva, 2014; Jain et al., 2020; Sayad et al., 2019), and have the potential to be employed in Earth system models to improve wildfire simulations (e.g., Zhu et al., 2022)."

Additionally, we discussed the potential for applying our approach to other ecosystems.

In L661,

"Overall, our study presents a reproducible approach that utilizes machine learning to identify parameter values that improve model fidelity against observations and promote coexistence between plant functional types in vegetation demography models across diverse ecosystems. This approach has the potential to enhance the modeling of PFT coexistence in various regions, such as the mixed conifer forests in Sierra Nevada, California (Buotte et al., 2021), Amazon forests subject to selective logging (Huang et al., 2020) and tropical forests with heterogeneous soils and subject to droughts in Panama (Cheng et al., 2021)."

---

## Author Response (AR2)

**Responses to comments <egusphere-2022-1286>**

Thank you very much for revising the manuscript and addressing all referee comments. Below are just a few minor formal issues that need to be addressed before publication.

We sincerely appreciate your valuable suggestions and the time you have taken to help improve our manuscript. Please find our responses to your specific comments provided below in blue.

Page 2: As the GMD manuscript composition does not provide for a list of highlights before the abstract, please remove the list from the main document.

We have removed the highlights.

Page 4: To avoid three different abstracts/summaries, I suggest aligning the "Plain Language Summary" with the "Short Summary" and removing this section from the main document.

We have aligned the "Plain Language Summary" with the "Short Summary", and removed this section from the main document.

Page 15, Fig. 2: In box P2, the "i" in "Xi" is not subscripted. Also, since the index i is not used elsewhere in the figure, I suggest omitting it and just using "X" and "Y".

Updated.

Page 16: Although you have added a reference to Table 1 for the parameter definitions, for better readability please put the variable names before the symbols, at least the first time they are mentioned, e.g.:

Line 294: "the maximum carboxylation rate $V\_cmax$"

Line 296: "leaf longevity $L\_leaf$ and wood density WD"

Line 297: "the specific leaf area SLA"

Thank you for your valuable suggestions. We have made the necessary updates to provide the full names of parameter symbols when they first appear in the text. Specifically, $M_{bk}$ in L270, $V_{cmax}$ in L271, $L_{leaf}$ in L273, $WD$ and $SLA$ in L274, and $CR_{s2l}$ in L282.

Page 18, line 327: Please expand "SHAP" here.

Since Section 2.5 already provides detailed explanations of SHAP, it would be repetitive to expand on SHAP here, and therefore, we removed the mention of SHAP here. Then, a slight modification is made to the last sentence to ensure clarity: "The ML models were trained, tested, and subsequently utilized to perform the parameter sensitivity analysis, as described in Section 2.5."

Page 28, line 496: "Figure 6" should read "Figure 7".

Modified.